# OpenReview forum: "An LLM can Fool Itself: A Prompt-Based Adversarial Attack"
_ICLR.cc/2024/Conference — ICLR 2024 poster_

### Official Review · Reviewer_tZSe · 2023-10-29

**Soundness:** 3 good
**Presentation:** 4 excellent
**Contribution:** 2 fair
**Rating:** 5
**Confidence:** 3

**Summary:**

This paper presents a new adversarial attack scheme against LLM using prompt engineering. The authors propose PromptAttack which includes three key components: original inputs, attack objective, and attack guidance. To enhance the attack efficacy, the authors also investigate the ensembling methods. Results show that PromptAttack can achieve high attack success rates (ASR) compared to AdvGLUE.

**Strengths:**

+ The authors show an effective adversarial attack against LLMs using prompt engineering. Particularly, PromptAttack designs fine-grained instructions to guide the victim LLM itself to generate adversarial samples that can fool itself.

+ The authors investigate the efficacy of PromptAttack using the few-shot strategy and ensembling strategy.

+ Empirical results show that PromptAttack achieves higher ASR on various benchmarks compared to AdvGLUE and AdvGLUE++.

**Weaknesses:**

The paper has the following weaknesses:
- The novelty of the proposed attack scheme is not clear. Although PromptAttack is shown to be effective, the working mechanism is straightforward and simple. It's not clear what is the challenge of designing an effective adversarial attack against LLMs.
- The contributions of the paper do not seem to be enough. The authors put together multiple existing techniques, including few-shot prompt engineering, ensembling, and adversarial attacks against the text. The contributions seem incremental and not substantial.

**Questions:**

Please consider addressing the weak points above.

---

> ### Author Response · Authors · 2023-11-14
> **Response to Review tZSe**
>
> Many thanks for your comments! Please find our replies below.
>
> > [Reply to Q1] No prior studies have studied how to ask the LLM to directly generate effective adversarial data via prompts. We are the first to propose a *non-trivial* attack prompt to effectively prompt the LLM to directly generate adversarial examples that can fool itself.
>
> So far, prompt engineering and in-context learning (e.g., few-shot inference) have been widely applied to solve various optimization tasks [1,2,3,4], but they have not yet been applied to utilize the LLM to solve the task of generating adversarial examples. We are the first to design an effective attack prompt that can elicit the victim LLM to output adversarial examples that can fool itself.
>
> Our design of the attack prompt template is non-trivial and reasonable. No prior studies have designed such attack prompts. We are the first to map the mathematical formulation of the conventional textual attacks (e.g., attack objective function and attack optimization method) into an attack prompt in a textual form. Empirical results validate the effectiveness of our attack prompt.
>
> > [Reply to Q2] We highlight our novelty and significant contributions as follows.
>
> 1. **Technically, PromptAttack introduces a novel paradigm of adversarial attack that prompts the LLM to directly generate the adversarial examples in a black-box manner, which exempts complex optimization procedures.**
>     - *Conventional adversarial textual attacks* require humans to propose complex optimization procedures (e.g., calculating prediction logits in a white-box manner [5] and computationally-expansive back-propagations [6]) to solve the attack objectives. PromptAttack, without complex optimization procedures, asks the LLM to directly generate adversarial examples via an attack prompt in a black-box manner, which is efficient and simple. We believe we opened a new research direction in studying how to let the LLM directly generate strong adversarial examples via attack prompts.
>     - *No prior studies in in-context learning (e.g., few-shot inference)* [1,2,3,4] have studied whether the LLM can solve the task of generating adversarial examples via prompts. We make the first effort to validate that the LLM can solve the task of generating adversarial examples via an attack prompt.
>     - *No prior studies in prompt engineering* have designed such attack prompt templates. We are the first to propose an effective attack prompt template inspired by conventional adversarial textual attacks.
>
> 2. **Empirically, PromptAttack is the state-of-the-art tool for evaluating the adversarial robustness of the (black-box) LLM in terms of effectiveness, adaptability, scalability, and efficiency.**
>    - *Effectiveness*. As shown in Table 2, PromptAttack achieves a significantly higher ASR compared to prior SOTA methods (AdvGLUE [8] and AdvGLUE++ [7]). It indicates that PromptAttack can discover more failure modes of existing (black-box) LLMs.
>    - *Adaptability*. Compared to transferable adversarial examples used in AdvGLUE and AdvGLUE++ that are fixed once generated, PromptAttack can adaptively generate the adversarial examples against the latest LLM via the attack prompt. In this way, PromptAttack can provide a reliable evaluation.
>    - *Scalability*. AdvGLUE and AdvGLUE++ use ensemble attacks which require white-box access to the LLM [5,6]. PromptAttack only requires querying the LLM in a black-box manner. Therefore, PromptAttack is applicable to various LLMs, even the black-box API (e.g., GPT-3.5).
>    - *Efficiency*. PromptAttack, without expensive computation (e.g., back-propagation), only requires black-box queries for each data points to generate an adversarial variant. We find that PromptAttack against GPT-3.5 consumes about 2 seconds to generate an adversarial variant for each data point, which validates the efficiency of PromptAttack.
>
> ---
>
> *References*
>
> [1] Chain-of-Thought Prompting Elicits Reasoning in Large Language Models, NeurIPS 2022.
>
> [2] language models are zero-shot reasoners, NeurIPS 2022.
>
> [3] Large language models are human-level prompt engineers, ICLR 2023.
>
> [4] Large Language Models as Optimizers, ICLR 2024 submission.
>
> [5] Bert-attack: Adversarial attack against bert using bert, EAAI 2020.
>
> [6] SemAttack: Natural Textual Attacks via Different Semantic Spaces, ACL 2022.
>
> [7] Decodingtrust: A comprehensive assessment of trustworthiness in gpt models, ArXiv 2023.
>
> [8] Adversarial glue: A multi-task benchmark for robustness evaluation of language models, NeurIPS (Dataset track) 2021.

---

> > ### Author Response · Authors · 2023-11-19
> > **We would like to know if you have any further questions or require additional clarification.**
> >
> > Dear **Reviewer tZSe**,
> >
> > Thank you again for your valuable comments! We have carefully considered your comments and have provided our responses.
> >
> > Please let us know if our replies have satisfactorily addressed your concerns. Please do not hesitate to let us know if you have any further questions or if you require any additional clarification.
> >
> > Thank you very much!
> >
> > Best wishes,
> >
> > Authors

---

> > > ### Author Response · Authors · 2023-11-21
> > > **Your feedback is critical to us.**
> > >
> > > Dear **Reviewer tZSe**,
> > >
> > > We were wondering if our responses have resolved your concerns since only two days are left for discussion. We highlight our significant contributions from the technical and empirical perspectives.
> > >
> > > Please let us know if our replies have satisfactorily addressed your concerns. We are eager to engage in further discussions and continue improving our work. Thank you very much!
> > >
> > > Best wishes,
> > >
> > > Authors of Paper #540

---

### Official Review · Reviewer_zRU1 · 2023-10-30

**Soundness:** 2 fair
**Presentation:** 3 good
**Contribution:** 2 fair
**Rating:** 6
**Confidence:** 5

**Summary:**

This paper focuses on adversarial attacks on large language models. To be more specific, this paper considers the black-box attacks and designs a structure of prompts to lead the model to generate adversarial prompts by themselves. Extensive experiments are conducted to illustrate the effectiveness of the proposed method.

**Strengths:**

The organization is good and the paper is easy to follow. The proposed method is simple and the effectiveness is promising via experiments.

**Weaknesses:**

1. Task description in section 4 is confusing. Please provide backgrounds in the appendix, showing what they are and why they matter.
2. This work is partially motivated by the lack of efficiency and effectiveness of existing adversarial attacks, but there is no illustration of efficiency in the experiment part.
3. The experiments are only conducted on 2 models, which is not enough, especially when Llama is an open-source model. I would recommend testing on more recent black-box models such as Bard, Claude, Palm.

**Questions:**

1. From Table 4, the ASR for each perturbation type is very low but the ASR in Table 3 is much higher (3-5 times higher). Why does this happen?
2. Could you provide some understandings of why PromptAttack works? Also, are there any defenses against adversarial attacks in LLM? If there exists, please evaluate those defenses.

---

> ### Author Response · Authors · 2023-11-14
> **Response to Review zRU1 (part 1)**
>
> Many thanks for your comments! Please find our replies below.
>
> > [Reply to W1] The task description is used for in-context learning, which illustrates the detailed objective of the optimization task in the form of natural language. We provide a detailed illustration in Appendix B.7 and provide examples of the task description for solving the SST-2 task in Table 20.
>
>
> > [Reply to W2] We demonstrate that PromptAttack is more computationally efficient in evaluating robustness of the LLM than AdvGLUE and AdvGLUE++.
>
> PromptAttack against GPT-3.5 only requires querying GPT-3.5 without any GPU resources, which consumes about 2 seconds to generate the adversarial example for each data point. In contrast, AdvGLUE and AdvGLUE++ consume much more running time and GPU memories to generate transferable adversarial examples since they require complex optimization procedures (e.g., calculating the predicted logits [1] and do back-propagations [2]) in their ensemble attacks against an ensemble of language models. The computational consumption is estimated on RTX A5000 GPUs.
>
> |Method| Need forwarding? | Need back-propagation? |Estimated consumption of the GPU memory|Estimated running time per data point|
> |-|-|-|-|-|
> | AdvGLUE | &#10003;| &#10003; | 16GB | 50 seconds |
> | AdvGLUE++ | &#10003;| &#10003; | 105GB | 330 seconds |
> | PromptAttack against GPT-3.5 | &#10003;| &#10005; | **- (via black-box API)** | **2 seconds** |
>
>
> > [Reply to W3] We provide the robustness evaluation of Palm 2 using PromptAttack in a black-box manner.
>
> The results show that GPT-3.5 is more adversarially robust than Palm 2 under PromptAttack.
>
> | Black-box LLM | ASR evaluated on SST-2 under PromptAttack |
> |-|-|
> | Google Palm 2 (version "text-bison-001") | 85.30 |
> | GPT-3.5 | 75.23 |
>
> Note that Google Bard is powered by Palm 2. We do not provide the results of Bard and Claude under PromptAttack since Google Bard API and Claude API are available to only a limited number of users [11,12]. Although we can access Bard and Claude via the web page, it would be impractical for us to test inputs one by one on a web page without the API.
>
> > [Reply to Q1] In Table 3, we utilize the ensemble strategy. In Table 4, we do not utilize the ensemble strategy.
>
> The ensemble strategy collects the adversarial examples generated by 9 different types of perturbation instructions together. Then, PromptAttack with the ensemble strategy (PromptAttack-EN) selects the adversarial example that can successfully fool the victim LLM as the output. In this way, PromptAttack-EN significantly improves the ASR.

---

> ### Author Response · Authors · 2023-11-14
> **Response to Review zRU1 (part 2)**
>
> > [Reply to Q2.1] We explain why PromptAttack works from the perspective of in-context learning.
>
> In-context learning [3,4,5,6,7,9,10] has been shown as a powerful method that asks an LLM to solve the optimization task via a well-designed prompt. The prompt describes the optimization task in the form of high-level natural language. Benefiting from the LLM's ability to understand natural language, the LLM can solve the optimization task [9].
>
> For example, [7] lets the LLM directly generate a good task instruction according to the prompt that aims to find a task instruction that optimizes the task accuracy. [9] lets the LLM directly generate a good solution for the mathematical question according to the prompt that aims to find a solution that optimizes the accuracy.
>
> Similarly, PromptAttack can be regarded as a novel type of in-context learning, which lets the LLM directly generate a new sentence (i.e., adversarial example) given the attack prompt that aims to find a sentence that fools itself.
> The LLM's powerful comprehension ability enables itself to solve the task of generating adversarial examples.
> It leaves future work to provide a deeper and theoretical understanding of why in-context learning works including [3,4,5,6,7,9,10] and our proposed PromptAttack.
>
>
> > [Reply to Q2.2] To the best of our knowledge, there is no defense against adversarial attacks in in-context learning. We try to utilize the superior in-context learning methods [4,5,6,7,10] and adversarial training (AT) [8] as defensive strategies and report the performance under the defense.
>
> First, we consider the superior in-context learning methods including few-shot in-context learning [4], chain-of-thought (CoT) [5,6], and automatic prompt engineering (APE) [7,10] as the defensive strategies. The performance is evaluated on GPT-3.5 using the SST-2 dataset.  The results show that the few-shot strategy is effective, but CoT and APE are not effective in defense.
>
> | Defense | ASR evaluated on SST-2 under PromptAttack |
> |-|-|
> | No defense (zero-shot in-context learning) | 76.98 |
> | Few-shot in-context learning | 71.62 |
> | CoT + Zero-shot in-context learning | 85.74 |
> | CoT + Few-shot in-context learning | 82.24 |
> | APE + Zero-shot in-context learning | 82.56 |
> | APE + Few-shot in-context learning | 80.68 |
>
> Second, although AT [8] is effective in defending against adversarial attacks, we cannot directly apply AT to LLMs due to computational prohibition. Thus, we conducted AT on BERT and RoBERTa, which validated the effectiveness of AT in robustifying language models.
>
> | Model | Standard BERT | Robust BERT | Standard RoBERTa | Robust RoBERTa |
> |-|-|-|-|-|
> | Average ASR over all tasks | 36.89 | **34.69** | 34.26 | **33.66** |
>
> The experimental results are copied from Table 16 in Appendix B.6. We used adversarial examples generated by PromptAttack against GPT-3.5 to evaluate the ASR on standardly and adversarially trained language models.
>
> ---
>
> **[Update more results under possible defense]**
>
> Concurrent works [13,14] are particularly designed for defending against jailbreak attacks, which is not applicable to defending against conventional adversarial attacks. Another concurrent work [15] uses the perplexity (PPL) filter to filter out low-quality prompts, thus defending against possible adversarial prompts, which is applicable to defending against adversarial examples. Here, we provide the ASR with or without the PPL filter to show that the PPL filter [15] is effective in defending against adversarial examples. PromptAttack is still a strong adversarial attack under this defense [15].
>
> | Method | without PPL filter (no defense) | with PPL filter [15] |
> |-|-|-|
> |AdvGLUE| 33.05 | 29.22|
> |AdvGLUE++| 4.95 |3.95|
> |PromptAttack-EN| 56.67|34.56|
> |PromptAttack-FS-EN|76.98|53.07|
>
> ---
>
> *References*
>
> [1] Bert-attack: Adversarial attack against bert using bert, EAAI 2020.
>
> [2] SemAttack: Natural Textual Attacks via Different Semantic Spaces, ACL 2022.
>
> [3] Pre-train, prompt, and predict: A systematic survey of prompting methods in natural language processing, ACM Computing Surveys, 2023.
>
> [4] What Can Transformers Learn In-Context? A Case Study of Simple Function Classes, ArXiv 2022.
>
> [5] Chain-of-Thought Prompting Elicits Reasoning in Large Language Models, NeurIPS 2022.
>
> [6] language models are zero-shot reasoners, NeurIPS 2022
>
> [7] Large language models are human-level prompt engineers, ICLR 2023.
>
> [8] Freelb: Enhanced adversarial training for natural language understanding, ICLR 2019.
>
> [9] Large Language Models as Optimizers, ICLR 2024 submission.
>
> [10] Use Your INSTINCT: INSTruction optimization usIng Neural bandits Coupled with Transformers, ICLR 2024 submission.
>
> [11] https://www.googlecloudcommunity.com/gc/AI-ML/Google-Bard-API/m-p/538517.
>
> [12] https://www.anthropic.com/earlyaccess
>
> [13] https://openreview.net/forum?id=xq7h9nfdY2
>
> [14] https://openreview.net/forum?id=wNere1lelo
>
> [15] https://openreview.net/forum?id=0VZP2Dr9KX

---

### Official Review · Reviewer_2JDp · 2023-11-01

**Soundness:** 2 fair
**Presentation:** 3 good
**Contribution:** 2 fair
**Rating:** 5
**Confidence:** 3

**Summary:**

The paper presents "PromptAttack", an adversarial attack method for evaluating the robustness of LLMs. It introduces a prompt-based adversarial attack that manipulates the victim LLM to generate adversarial samples by itself. The attack is composed of three elements: original input, attack objective, and attack guidance. A fidelity filter is employed to ensure that the adversarial samples maintain semantic meaning. The paper evaluates PromptAttack on Llama2 and GPT-3.5 that outperforms existing benchmarks like AdvGLUE and AdvGLUE++ in ASR. It raises important questions about the reliability and safety of deploying LLMs in critical applications.

**Strengths:**

1. The paper is easy to follow.
2. The study of adversarial attacks in LLM is important and interesting.

**Weaknesses:**

1. The paper methodology technical approach lacks novelty, which is essentially an application of known techniques, e.g. the design of the perturbation instruction, fidelity filter, few-shot inference and ensemble attack. Can the authors explain more what is the unique contribution?
2. As the authors mentioned, the scale of LLMs may impact attack performance. If so, a more comprehensive evaluation of PromptAttack across a range of LLM scales, along with an analysis of computational overhead, would strengthen the paper.
3. The paper would benefit from a discussion on potential countermeasures or mitigation strategies that could enhance the robustness of LLMs against such attacks like PromptAttack.

**Questions:**

Please refer to the weaknesses.

---

> ### Author Response · Authors · 2023-11-14
> **Response to Reviewer 2JDp (part 1)**
>
> Many thanks for your comments! Please find our replies below.
>
> > [Reply to Q1] We highlight our novelty and significant contributions as follows:
>
> 1. **Technically, PromptAttack introduces a novel paradigm of adversarial attack that prompts the LLM to directly generate the adversarial examples in a black-box manner, which exempts complex optimization procedures.**
>     - *Conventional adversarial textual attacks* require humans to propose complex optimization procedures (e.g., calculating prediction logits in a white-box manner [1] and computationally expansive back-propagations [2]) to solve the attack objectives. PromptAttack, without complex optimization procedures, asks the LLM to directly generate adversarial examples via an attack prompt in a black-box manner, which is efficient and simple. We believe we opened a new research direction in studying how to let the LLM directly generate strong adversarial examples via attack prompts.
>     - *No prior studies in in-context learning (e.g., few-shot inference)* [4,5,6,7,10] have studied whether the LLM can solve the task of generating adversarial examples via prompts. We make the first effort to validate that the LLM can solve the task of generating adversarial examples via an attack prompt.
>     - *No prior studies in prompt engineering* have designed such attack prompt templates. We are the first to propose an effective attack prompt template inspired by conventional adversarial textual attacks.
>
> 2. **Empirically, PromptAttack is the state-of-the-art tool for evaluating the adversarial robustness of the (black-box) LLM in terms of effectiveness, adaptability, scalability, and efficiency.**
>    - *Effectiveness*. As shown in Table 2, PromptAttack achieves a significantly higher ASR compared to prior SOTA methods (AdvGLUE [9] and AdvGLUE++ [3]). It indicates that PromptAttack can discover more failure modes of existing (black-box) LLMs.
>    - *Adaptability*. Compared to transferable adversarial examples used in AdvGLUE and AdvGLUE++ that are fixed once generated, PromptAttack can adaptively generate the adversarial examples against the latest LLM via the attack prompt. In this way, PromptAttack can provide a reliable evaluation.
>    - *Scalability*. AdvGLUE and AdvGLUE++ use ensemble attacks which require white-box access to the LLM [1,2]. PromptAttack only requires querying the LLM in a black-box manner. Therefore, PromptAttack is applicable to various LLMs, even the black-box API (e.g., GPT-3.5).
>    - *Efficiency*. PromptAttack, without expensive computation (e.g., back-propagation), only requires black-box queries for each data point to generate an adversarial variant. We find that PromptAttack against GPT-3.5 consumes about 2 seconds to generate an adversarial variant for each data point, which validates the efficiency of PromptAttack.
>
>
> > [Reply to Q2] We provide an analysis on the scale of the LLM w.r.t. the attack performance and computational overhead as follows.
>
> |Victim LLM|Number of parameters|ASR under PromptAttack| Estimated running time per data point| Estimated consumption of the GPU memory|
> |-|-|-|-|-|
> | Llama2-7B | 7B | 50.82 | 10 seconds | 14GB (using RTX A5000 GPUs) |
> | Llama2-13B | 13B | 63.20 | 20 seconds | 27GB (using RTX A5000 GPUs) |
> | GPT-3.5 | 154B$^*$ | 48.34 | 2 seconds | - (via black-box API) |
>
> $^*$The number of parameters of GPT-3.5 is provided by [this website](https://www.ankursnewsletter.com/p/gpt-4-gpt-3-and-gpt-35-turbo-a-review), which could be inaccurate since OpenAI does not publish the accurate number of parameters of GPT-3.5.
>
> 1. As the scale of the LLM (measured as the number of model parameters) increases, the ASR of PromptAttack first increases and then decreases.
>
>    The increase of ASR from Llama2-7B to Llama2-13B could be owing to that the LLM’s comprehension ability increases as the model becomes larger, which increases the quality of generated adversarial examples (see detailed analysis in Section 4.1). The decrease of ASR from Llama2-13B to GPT-3.5 could be owing to that GPT-3.5 has more parameters, which leads to stronger robustness.
>
> 2. PromptAttack is much more computationally efficient than AdvGLUE and AdvGLUE++. We show that AdvGLUE and AdvGLUE++ require much more computational resources to generate transferable adversarial examples in the following table.
>
> |Method|Estimated running time per data point| Estimated consumption of the GPU memory (using RTX A5000 GPUs)|
> |-|-|-|
> | AdvGLUE | 50 seconds | 16GB |
> | AdvGLUE++ | 330 seconds | 105GB |
> |PromptAttack against GPT-3.5| 2 seconds | - (via black-box API) |

---

> ### Author Response · Authors · 2023-11-14
> **Response to Reviewer 2JDp (part 2)**
>
> > [Reply to Q3] To the best of our knowledge, there is no defense against adversarial attacks in in-context learning. We try to utilize the superior in-context learning methods [4,5,6,7,10] and adversarial training (AT) [8] as defensive strategies and report the performance under the defense.
>
> First, we consider the in-context learning methods including few-shot in-context learning [4], chain-of-thought (CoT) [5,6], and automatic prompt engineering (APE) [7,10] as the defensive strategies. The performance is evaluated on GPT-3.5 using the SST-2 dataset.  The results show that the few-shot strategy is effective, but CoT and APE are not effective in defense.
>
> | Defense | ASR evaluated on SST-2 under PromptAttack |
> |-|-|
> | No defense (zero-shot in-context learning) | 76.98 |
> | Few-shot in-context learning | 71.62 |
> | CoT + Zero-shot in-context learning | 85.74 |
> | CoT + Few-shot in-context learning | 82.24 |
> | APE + Zero-shot in-context learning | 82.56 |
> | APE + Few-shot in-context learning | 80.68 |
>
> Second, although AT [8] is effective in defending against adversarial attacks, we cannot directly apply AT to LLMs due to computational prohibition. Thus, we conducted AT on BERT and RoBERTa, which validates the effectiveness of AT in robustifying language models.
>
> | Model | Standard BERT | Robust BERT | Standard RoBERTa | Robust RoBERTa |
> |-|-|-|-|-|
> | Average ASR over all tasks | 36.89 | **34.69** | 34.26 | **33.66** |
>
> The experimental results are copied from Table 16 in Appendix B.6. We used adversarial examples generated by PromptAttack against GPT-3.5 to evaluate the ASR on standardly and adversarially trained language models.
>
> ---
>
> **[Update more results under possible defense]**
>
> There are three concurrent works [13,14,15] that aim to defend against adversarial prompts. [13,14] particularly designed for defending against jailbreak attacks, which is not applicable to defending against conventional adversarial attacks. We utilized the defensive strategy in [15] which uses the perplexity (PPL) filter to filter out low-quality prompts, thus defending against possible adversarial prompts. Here, we provide the ASR with or without the PPL filter to show that the PPL filter [15] is effective in defending against adversarial examples. The results further validate that our PromptAttack is still a strong adversarial attack under this defensive strategy [15].
>
> | Method | without PPL filter (no defense) | with PPL filter [15] |
> |-|-|-|
> |AdvGLUE| 33.05 | 29.22|
> |AdvGLUE++| 4.95 |3.95|
> |PromptAttack-EN| 56.67|34.56|
> |PromptAttack-FS-EN|76.98|53.07|
>
> ---
>
> *References*
>
> [1] Bert-attack: Adversarial attack against bert using bert, EAAI 2020.
>
> [2] SemAttack: Natural Textual Attacks via Different Semantic Spaces, ACL 2022.
>
> [3] Decodingtrust: A comprehensive assessment of trustworthiness in gpt models, ArXiv 2023.
>
> [4] What Can Transformers Learn In-Context? A Case Study of Simple Function Classes, ArXiv 2022.
>
> [5] Chain-of-Thought Prompting Elicits Reasoning in Large Language Models, NeurIPS 2022.
>
> [6] language models are zero-shot reasoners, NeurIPS 2022
>
> [7] Large language models are human-level prompt engineers, ICLR 2023.
>
> [8] Freelb: Enhanced adversarial training for natural language understanding, ICLR 2019.
>
> [9] Adversarial glue: A multi-task benchmark for robustness evaluation of language models, NeurIPS (Dataset track) 2021.
>
> [10] Use Your INSTINCT: INSTruction optimization usIng Neural bandits Coupled with Transformers, ICLR 2024 submission.
>
> [11] https://blog.research.google/2022/04/pathways-language-model-palm-scaling-to.html
>
> [12] https://www.cnbc.com/2023/05/16/googles-palm-2-uses-nearly-five-times-more-text-data-than-predecessor.html
>
> [13] https://openreview.net/forum?id=xq7h9nfdY2
>
> [14] https://openreview.net/forum?id=wNere1lelo
>
> [15] https://openreview.net/forum?id=0VZP2Dr9KX

---

> > ### Author Response · Authors · 2023-11-19
> > **We would like to know if you have any further questions or require additional clarification.**
> >
> > Dear **Reviewer 2JDp**,
> >
> > Thank you again for your valuable comments! We have carefully considered your comments and have provided our responses.
> >
> > Please let us know if our replies have satisfactorily addressed your concerns. Please do not hesitate to let us know if you have any further questions or if you require any additional clarification.
> >
> > Thank you very much!
> >
> > Best wishes,
> >
> > Authors

---

> > > ### Author Response · Authors · 2023-11-21
> > > **Your feedback is critical to us.**
> > >
> > > Dear **Reviewer 2JDp**,
> > >
> > > We were wondering if our responses have resolved your concerns since only two days are left for discussion. We highlight our unique contributions from the technical and empirical perspectives. Besides, by carefully considering your comments, we have provided additional empirical analyses.
> > >
> > > Please let us know if our replies have satisfactorily addressed your concerns. We are eager to engage in further discussions and continue improving our work. Thank you very much!
> > >
> > > Best wishes,
> > >
> > > Authors of Paper #540

---

### Author Response · Authors · 2023-11-18
**We would like to know if you have any further questions or require additional clarification.**

Dear Reviewers,

Thank you for taking the time to review our work and for providing us with valuable feedback. We have carefully considered your comments and have provided our responses.

If you have any further questions or require additional clarification, please kindly let us know.

Thank you again for your valuable input.

Best wishes,

Authors

---

### Author Response · Authors · 2023-11-23
**A summary of rebuttal.**

Dear All Reviewers:

Thank you again for taking the time to review our work and for providing us with valuable feedback.

Here is a summary of our rebuttal:

1. **Novelty and contributions** are highlighted as follows:
   - **Technically**, PromptAttack introduces a novel paradigm of adversarial attack that prompts the LLM to directly generate the adversarial examples in a black-box manner, which exempts complex optimization procedures.
   - **Empirically**, PromptAttack is the state-of-the-art tool for evaluating the adversarial robustness of the (black-box) LLM in terms of effectiveness, adaptability, scalability, and efficiency.

2. **Efficiency of PromptAttack**. We empirically validate that PromptAttack is much more computationally efficient compared to AdvGLUE and AdvGLUE++. Notably, PromptAttack against GPT-3.5 does not require any GPU memories and only consumes about 2 seconds per data point.

3. **Defense**. We provide a comprehensive analysis of the attack performance under possible defensive strategies. We find that few-shot inference, adversarial training, and the perplexity filter are effective defensive strategies while chain-of-thought and automatic prompt tuning are not. PromptAttack consistently achieves the highest ASR under various defensive methods.

Today is the last day of the discussion period. If you have a chance, please read through our rebuttal and let us know whether we have addressed your concerns. We greatly appreciate your ongoing engagement with our paper and the feedback you have provided.

Best wishes,

Authors

---

### Meta-Review · Area_Chair_roeQ · 2023-12-20

**Metareview:**

This paper focuses on adversarial attacks on LLMs, and proposes an efficient tool to audit the LLM’s adversarial robustness via a prompt-based adversarial attack (PromptAttack). PromptAttack is a prompt-based adversarial attack that manipulates the victim LLM to generate adversarial samples by itself. The attack is composed of three elements: the original input, the attack objective, and the attack guidance. A fidelity filter is employed to ensure that the adversarial samples maintain semantic meaning. The paper evaluates PromptAttack on Llama2 and GPT-3.5 and reports that the proposed method outperforms existing benchmarks like AdvGLUE and AdvGLUE++ in ASR.

This is a borderline paper. Unfortunately, the reviewers were not responsive despite my efforts. I further asked a different person (an expert in the field) about their opinion on the paper (which was positive). All in all, I think that the paper has sufficient novelty to be accepted. I strongly recommend that the authors incorporate all the comments from the reviewers in the revised manuscript.

**Justification For Why Not Higher Score:**

See above

**Justification For Why Not Lower Score:**

See above

---

### Decision · Program_Chairs · 2024-01-16

Accept (poster)